# Diindolylmethane Inhibits Cadmium-Induced Autophagic Cell Death via Regulation of Oxidative Stress in HEL299 Human Lung Fibroblasts

**DOI:** 10.3390/molecules27165215

**Published:** 2022-08-16

**Authors:** Yeon-Seop Jung, Ho Jeong Lee, Moonjung Hyun, Hye-Jin Kim, Je-Hein Kim, Kwang-Hyun Hwang, Woong-Soo Kim, Jungil Choi, Jeong Doo Heo

**Affiliations:** 1Preclinical Research Center, Daegu Gyeongbuk Medical Innovation Foundation, Daegu 41061, Korea; 2Gyeongnam Bio-Health Research Support Center, Gyeongnam Branch Institute, Korea Institute of Toxicology (KIT), 17 Jeigok-gil, Jinju 52834, Korea

**Keywords:** cadmium, cytotoxicity, oxidative stress, diindolylmethane, antioxidant, fibroblast

## Abstract

Cadmium (Cd), a harmful heavy metal, can lead to various pulmonary diseases, including chronic obstructive pulmonary disease (COPD), by inducing cytotoxicity and disturbing redox homeostasis. The aim of the present study was to investigate Cd-mediated cytotoxicity using human lung fibroblasts and the therapeutic potential of 3,3′-diindolylmethane (DIM). Cadmium significantly reduced the cell viability of human embryonic lung (HEL299) cells accompanied by enhanced oxidative stress as evidenced by the increased expression of autophagy-related proteins such as LC3B and p62. However, treatment with DIM significantly suppressed autophagic cell death in Cd-induced HEL299 fibroblasts. In addition, DIM induced antioxidant enzyme activity and decreased intracellular reactive oxygen species (ROS) levels in Cd-damaged HEL299 cells. This study suggests that DIM effectively suppressed Cd-induced lung fibroblast cell death through the upregulation of antioxidant systems and represents a potential agent for the prevention of various diseases related to Cd exposure.

## 1. Introduction

Cadmium (Cd) is widely used in rechargeable nickel–cadmium batteries, pigments, coatings, and electroplating; however, it is a serious environmental toxicant and harmful to human health, and can cause diseases including chronic obstructive pulmonary disease (COPD) and fibrosis [1,2,3]. In addition, a number of investigators reported that acute exposure to high concentrations of Cd might lead to cell death, with an association noted between Cd-mediated autophagy–lysosomal degradation and cellular organelle homeostasis in various cell types [4,5,6].

There is increasing evidence supporting the findings that excessive Cd exposure markedly elevates and mediates reactive oxygen species (ROS) levels, including superoxide radicals, hydroxyl radicals, as well as nitric oxides as contributors to adverse cellular homeostasis regulation. This phenomenon is also known to be involved in autophagic cell death by inducing GFP-LC3B puncta and the LC3B-II transformation as autophagy markers in human lung bronchial epithelial cells and neuronal cells [4,7,8]. Several investigators reported that plant extracts and natural and synthetic antioxidants, such as glutathione (GSH), N-acetyl cysteine (NAC), vitamin C, and *Allium cepa* L., inhibited Cd-mediated oxidative stress and improved the intracellular redox system, which resulted in the suppression of oxidative stress-related cell death [9,10]. Based upon these findings from several different labs, the evidence indicated that acute Cd exposure mediates autophagy by disrupting redox homeostasis, and actions of various antioxidants constitute a potential preventive and therapeutic strategy for Cd-mediated cytotoxicity.

The indole-3-carbinol and its metabolite 3,3′-diindolylmethane (DIM) are present in cruciferous vegetables such as broccoli and cauliflower [11]. Further, these compounds act as estrogen receptor antagonists and are considered as preventive agents [12]. Among the indole-3-carbinol compounds, DIM has also been used as a food supplement to treat various cancers [13,14,15], endometriosis [16], premenstrual syndrome [17], and weight loss [18]. In addition, several investigators reported that DIM exerted multiple biological activities such as antiangiogenic [19], antiproliferative [15], neuroprotective [20], and antioxidant activities [14,19,21]. However, little information is currently available regarding DIM’s potential to protect against the Cd-mediated disruption of redox homeostasis and autophagic cell death in human normal lung fibroblasts. Therefore, it was postulated that DIM might influence Cd-mediated cytotoxicity. The aim of this study was to examine the influence of cadmium chloride (CdCl_2_) on cell viability in part by affecting autophagy and, subsequently, disrupting the intracellular redox systems. Then, preventive effects of DIM were investigated in Cd-induced human embryonic lung (HEL) 299 normal lung fibroblasts.

## 2. Results

### 2.1. Cadmium Chloride (Cd) Induces Autophagic Cell Death in HEL299 Cell Lines

Prior to the investigation of high-concentration Cd-exposure-mediated toxic effects on normal human lung fibroblasts, the concentration dependence of cytotoxic effects of Cd by means of an MTT assay in HEL299 cells treated for 24 h and 48 h with different concentration of Cd was investigated. As illustrated in Figure 1A, treatment with Cd at higher concentrations (75 and 100 μM) significantly decreased the viability of HEL299 cells (growth-inhibitory IC50 values were 80.94 μM, 24 h), but not in cells treated with a low concentration (25 and 50 μM). Further, the cytotoxicity of Cd was established using the microscopy of HEL299 cells exposed for 24 h with different amounts of Cd. Cells incubated with Cd at concentrations higher than 75 μM exhibited significant morphology changes, including the appearance of segments of cell bodies (Appendix A). To investigate whether Cd-induced lung fibroblast cell death was due to an activated apoptosis pathway, a flow cytometry assay was performed using cells stained with annexin V and propidium Iodide (PI) solution. Treatment with Cd at higher concentrations (75 and 100 μM) markedly increased the PI-positive cell population accompanied by minimal annexin V staining (Figure 1B,C). As illustrated in Appendix A, the protein expression of apoptosis markers (caspase 3, caspase 9, PARP, and Bax) was significantly decreased via treatment with Cd, suggesting that acute Cd exposure contributes to an alternative type of cell death such as autophagy.

Therefore, we tested the expression of autophagy-related proteins after Cd treatment. As shown in Figure 1D,E, the protein expression of LC3B and AMPKα phosphorylation was significantly increased compared to controls after Cd treatment, whereas p62 protein expression was decreased by treatment with Cd in a concentration-dependent manner. Bafilomycin A1 (BFM), an autophagy flux inhibitor, significantly restored Cd-induced reduced cell viability (Appendix A). These results indicated that HEL299 cells were damaged 24 h after treatment with Cd by autophagy induction.

### 2.2. Cadmium Induces Oxidative Stress in Lung Fibroblast

In order to understand the mechanisms underlying Cd-mediated autophagic cell death, cellular ROS levels in Cd-treated cells were measured using the DCF-DA probe. Figure 2A shows that HEL299 cells exposed to Cd at 50 μM did not appear to markedly affect ROS levels; however, treatment with Cd at higher concentrations (75 and 100 μM) significantly elevated ROS levels, which was confirmed with a fluorescence microscopy analysis (Figure 2B).

Previous studies revealed that protein expressions of antioxidant-related genes (heme oxygenase-1 (HO-1), catalase (CAT), and Nrf2)) are involved in cellular redox homeostasis through the suppression of intracellular ROS production, which is closely associated with endoplasmic reticulum (ER) stress [1,22]. To determine the activation of Nrf2 in HEL299 cells, we performed a Western blot analysis using nuclear fraction after Cd treatment. Figure 2C shows that Nrf2 expression significantly increased after Cd treatment in a dose-dependent manner. As presented in Figure 2D, treatment with Cd at higher concentrations (75 and 100 μM) significantly decreased the protein expression of CAT, while ER stress markers (GRP78 and CHOP) were increased by exposure to Cd. HEL299 cells incubated with Cd displayed high protein expression levels of Nrf2, Keap1, and its downstream target, HO-1, and all of these proteins were attenuated after coincubation with N-acetylcysteine (NAC, a ROS inhibitor) or azoramide (AZO, an ER stress inhibitor). Treatment with NAC at 5 mM markedly rescued lowered cell viability from Cd-mediated cell death (data not shown), and also reduced ROS production (Figure 2E). However, HO-1 protein expression was reduced after NAC exposure (Figure 2F). Data, thus, demonstrate that Cd-mediated cytotoxicity was predominantly associated with oxidative stress, and among several antioxidants, CAT was proposed as a potential candidate to be used for the prevention of Cd-initiated toxicity.

### 2.3. Diindolylmethane (DIM) Inhibits Cd-Mediated Autophagic Cell Death of HEL299 Cells

To further elucidate the pharmacological potential of DIM on Cd-mediated cell viability, an MTT assay, Western blot analysis, and microscopy were employed. DIM at concentrations lower than 60 μM did not exert a significant cytotoxic effect on cells, but an 80% decrease in cell viability was noted at 100 μM (Figure 3A), and the Cd-mediated fibroblast cotreatment with DIM for 24 h exhibited significant protective effects in a concentration-dependent manner (Figure 3B). The segmentation of cell bodies was determined with microscopy in Cd-exposed HEL299 cells. Treatment with DIM at 20 and 40 μM markedly altered cell morphology (Figure 3C). In the present study, acute Cd exposure clearly induced autophagic cell death. As shown in Figure 3D,E, treatment with DIM at 20–40 μM restored the protein expression levels of LC3B and p62. Together, these results indicate that DIM induced autophagic cell death and suppressed Cd-stimulated cytotoxicity.

### 2.4. DIM Inhibits Cd-Mediated Autophagic Cell Death by Regulating the Intracellular ROS and ER Stress Levels

Previously, cells treated with Cd at 75 and 100 μM for 24 h exhibited significantly increased intracellular ROS levels, while incubation with NAC (ROS inhibitor) or AZO (ER stress inhibitor) markedly restored Cd-mediated autophagic cell death in HEL299 cells. As illustrated in Figure 4A,B, the cellular ROS levels of HEL299 cells were increased by treatment with Cd compared to controls. However, incubation with DIM at 20–40 μM inhibited the Cd-mediated ROS production in a concentration-dependent manner, and this effect was also confirmed using fluorescence microscopy. Further, incubation with DIM markedly reduced Cd-induced protein levels of ER stress markers (GRP78 and CHOP) and antioxidant enzymes (HO-1), while CAT protein expression was increased by treatment with DIM (Figure 4C,D). These data suggest that DIM treatment as an antioxidant suppressed elevated intracellular ROS levels and cytotoxicity against acute Cd exposure in human lung fibroblasts.

## 3. Discussion

In recent years, a number of investigators demonstrated that autophagy is a catabolic process that enables cells to break down intracellular organelles, such as ribosomes and mitochondria, and supports cell survival mechanisms against various stressful environments [23,24]. However, cell death may result in the excessive self-digestion and degradation of essential cellular constituents [25,26]. The autophagic cell death is accompanied by the increased expression of LC3B and decreased expression of p62, which are both controlled by AMP-activated kinase (AMPK) signaling, and these proteins are important indicators in autophagic flux [27]. Moreover, Son et al. (2011) reported that Cd induced autophagic cell death by modulating ROS-dependent liver kinase B1 (LKB1)-AMPK signaling and its downstream target genes in JB6 mouse epidermal cells. The authors of [28] clearly demonstrated that using a GFP-LC3 construct that significantly increased GFP-LC3 puncta during autophagosome formation supports the typical induction of autophagy, and our results also confirmed an increase in autophagy cell death and protein expression (LC3B) after Cd exposure (Figure 1). In fact, p62 is known to be modulated by transcriptional regulation and post-translational autophagic degradation, which is induced by oxidative stress (Nrf2-Keap1 pathway) and some chemical compounds. Under oxidative stress, Nrf2 is released from the Keap1–Nrf2 complex through p62 binding to Keap1 competitively, and then is constantly degraded by autophagy [29]. Like previous reports, Cd-stimulated cells showed the generation of significant oxidative stress, and also induced Nrf2 protein expression (Figure 2C). Moreover, the protein expression of p62 was significantly reduced in Cd-stimulated cells, which was thought to be due to the acts as a substrate during autophagic degradation. Thus, attention has recently been focused on the role of antioxidants in Cd-induced cell death and the inhibitory effects of ROS production, either directly or indirectly.

DIM, 3,3′-diindolylmethane, is a major derivative and unique dimer of indole-3-carbinol (I3C) among many oligomeric products which exhibit antiproliferation and anticancer activities in different cancer cells, including prostate, breast, endometrial, colorectal, and pancreatic cancer [13,14,15,20,30]. The authors of found that DIM suppressed the 2,4,6-trinitrobenzene sulfonic acid (TNBS)-induced colitis in an animal model by reducing ROS generation, as well as diminishing the expression of vascular cell adhesion molecule 1 (VCAM1), which is typically enhanced by ROS. Keap1–Nrf2 signaling might also participate in DIM-reduced oxidative stress [31]. Several investigators noted that the Keap1–Nrf2 signaling pathway regulates the cellular defense mechanisms against oxidative stress [32]. However, autophagy is also directly involved in the degradation of p62 and Keap1 in a p62 interaction- and aggregation-dependent manner, and the subsequent activation of Nrf2 translocation [33,34]. Treatment with DIM at a concentration range which did not exhibit significant cytotoxicity significantly blocked ROS generation and regulated Keap1–Nrf2 signaling and its downstream target (HO-1) in Cd-stimulated HEL299 cells (Figure 4A,C). Exposure to DIM effectively protected cells from Cd-mediated cell death, and this effect was confirmed with microscopy and Western blot analysis (Appendix A). Previously, DIM was reported to reduce oxidative stress, stimulate the expression of antioxidant response element-mediated proteins, and protect against DNA damage via its antioxidant activity in normal human mammary epithelial cells and mouse embryonic fibroblasts [35,36]. The data indicate that DIM protected Cd-mediated cytotoxicity via the maintenance of the cellular redox balance in HEL299 cells.

In summary, the results obtained from this study showed that the selective induction of oxidative stress by Cd produced autophagic cell death. DIM significantly inhibited the Cd-mediated cytotoxicity of human lung fibroblasts. Data suggest that the inhibition of cell death by DIM correlated highly with the regulation of redox systems.

## 4. Materials and Methods

### 4.1. Antibodies, Reagents, and Chemicals

The antibodies used in this study are shown in Appendix A. Cadmium chloride (CdCl2, #202908), 3,3′-diindolylmethane (DIM, #D9568), N-Acetyl-L-cysteine (NAC, #A7250), Azoramide (AZO, #SML-1560), and all other chemicals were purchased from Sigma Aldrich (St. Louis, MO, USA) unless otherwise indicated. CdCl2 (Cd) was dissolved in sterile water at a concentration of 100 mM and was diluted in cell culture media to reach the indicated concentration.

### 4.2. Cell Culture and Treatment

Human embryonic lung (HEL) 299 cells were purchased from the American Type Culture Collection (Rockville, MD, USA) and grown in DMEM (Gibco-BRL, Rockville, MD, USA) supplemented with 10% fetal calf serum (FBS) and 1% antibiotics (penicillin and streptomycin), and incubated at 37 °C in a humidified chamber with 5% CO_2_. Once the cells reached approximately 80% confluence, they were harvested and subcultured after trypsin/EDTA treatment. HEL299 cells were treated with different concentrations of CdCl2 (25, 50, 75, or 100 μM) or DIM (5, 10, 20, 40, 60, 80, or 100 μM) for 24 h, and treatment with 20 and 40 μM DIM for 24h in the presence or absence of CdCl2 (100 μM) was used for further experiments.

### 4.3. Cell Viability Assay

The HEL299 cells (2 × 10^4^ cells/well) were allowed to adhere for at least 24 h before exposure. Following adherence, growth media were removed and replaced with assay media containing the appropriate treatments. Next, the medium was replaced with MTT (3-(4,5-dimethylthiazol-2yl)-2,5-diphenyltetrazolium bromide) solution (2 mg/mL) and incubated at 37 °C in a cell culture incubator for 3 h. The formazan crystals formed by MTT reduction were dissolved with DMSO and measured at 540 nm with a microplate reader (BioTek, Winooski, VT, USA).

### 4.4. Western Blot Analysis

Western blot analysis was undertaken as previously described (Lee et al., 2006). Whole-cell lysates were prepared in RIPA lysis buffer (Thermos Scientific, Waltham, MA, USA) supplemented with protease and phosphatase inhibitor cocktails (Gendepot, Katy, TX, USA). Total cell proteins were size-fractionated using SDS-PAGE and electro-transferred to Immobilon-P membranes (Millipore Corp., Bredford, MA, USA). Detection of specific proteins was carried out with enhanced chemiluminescence, following the manufacturer’s instructions (Amersham Biosciences, Piscataway, NJ, USA).

### 4.5. Measurement of Intracellular Level of ROS

Intracellular ROS levels were measured with flow cytometry using the peroxide-sensitive fluorescent probe 2′7′-dichlorofluorescin diacetate (DCF-DA). Briefly, HEL299 cells (3.5 × 106 cells/well) were plated in 6-well culture plates and exposed to each compound for 6 h. Cells were then incubated with DCF-DA (25 μM) in PBS at 37 °C for 30 min, washed twice with PBS, and detached using treatment with trypsin-EDTA. The detached cells were collected and resuspended in PBS, and fluorescence intensity of cells was measured using Muse Cell Analyzer (Merck, Millipore, Burlington, MA, USA). ROS levels were also measured by using fluorescence microscopy as described previously [37].

### 4.6. Annexin V and Dead Cell Analysis

Annexin V and dead cells were assayed using the multifunctional Muse Annexin V and Dead Cell kit (Millipore, Billerica, MA, USA), according to the user’s guide and manufacturer’s instructions. Briefly, HEL299 cells were harvested and washed twice with serum-free culture medium. Then, 5 μL FITC-labeled enhanced annexin V and 5 μL (20 μg/mL) propidium iodide were added to a 100 μL cell suspension. After incubation, in the dark, for 15 min at room temperature, samples were immediately analyzed using Muse Cell Analyzer (Merck, Millipore).

### 4.7. Statistical Analysis

Statistical significance of differences between groups was analyzed using one-way ANOVA with Dunnett’s multiple test (Prism 5, GraphPad, San Diego, CA, USA). The results were expressed as means with SE for 3 experiments for each group, unless otherwise indicated, and a *p* value of less than 0.05 was considered statistically significant.

## 5. Conclusions

DIM and related compounds were proposed as potential candidates for use in preventing oxidative stress related to cell death in fibroblasts. Our ongoing studies are focused on investigating the mechanisms underlying DIM-mediated fibroblast cell protective effects and developing other novel natural compounds capable of reducing cellular oxidative stress.

## Figures and Tables

**Figure 1 molecules-27-05215-f001:**
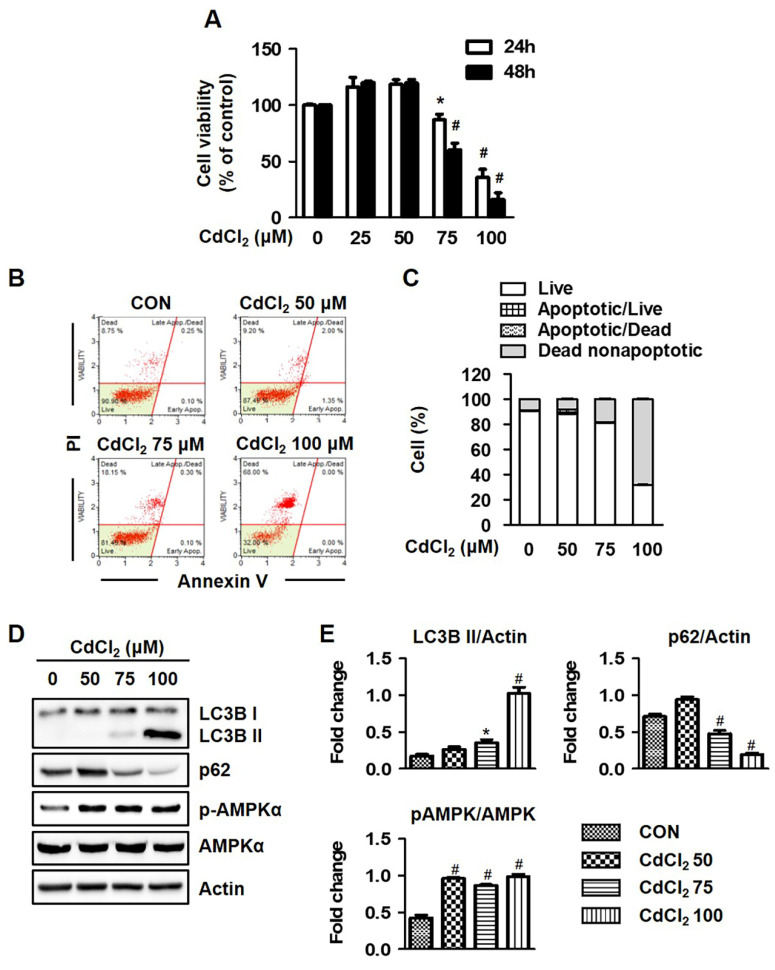
Cadmium induced autophagic cell death in HEL299 cells. (**A**) Cells were treated with the indicated concentration of CdCl_2_ for 24 and 48 h, and cell viability was measured using an MTT assay as described in the Material and Methods Section. (**B**,**C**) Cells were treated with different concentrations of CdCl_2_ for 24 h, and dead cells were determined by using Muse Cell Analyzer with Annexin V staining. (**D**,**E**) Cells were treated with the indicated concentration of CdCl_2_ for 24 h, and whole-cell lysates were analyzed with Western blot analysis. Actin was used as a loading control. All results are presented as means ± SE of three experiments. * *p* < 0.05 vs. control; # *p* < 0.001 vs. control.

**Figure 2 molecules-27-05215-f002:**
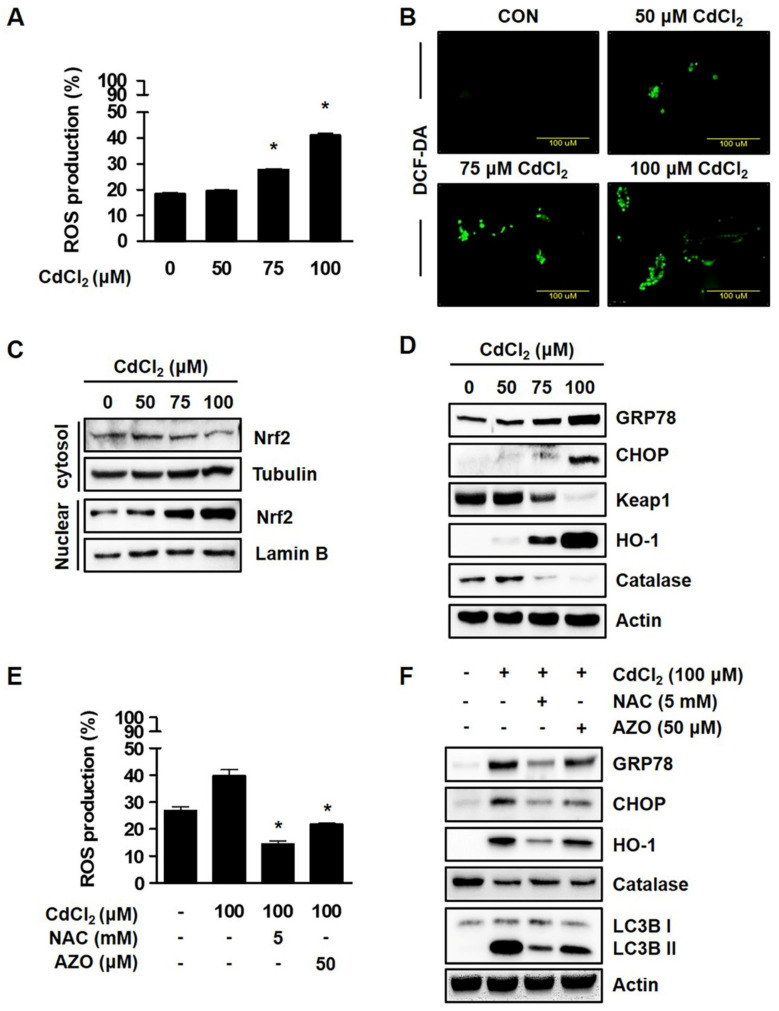
Cadmium induced oxidative stress in lung fibroblasts. (**A**,**B**) Cells were treated with the indicated concentration of CdCl_2_ for 24 h, and ROS production was determined using Muse Cell Analyzer with Muse Oxidative Stress Reagent as described in the Materials and Methods Section (**A**) and fluorescence microscopy (×100 magnification) (**B**). (**C**) Cytosolic and nuclear extracts were isolated after different concentrations of CdCl_2_ treatment for 6 h. Nrf2 expression was normalized to the cytosolic loading control, tubulin, and nuclear loading control, lamin B. (**D**) Cells were treated with different concentrations of CdCl_2_ for 24 h, and whole-cell lysates were analyzed with Western blot analysis using indicated antibodies. (**E**,**F**) Cells were pretreated with 5 mM NAC (antioxidant), and 50 µM AZO (ER stress inhibitor) in presence of 100 µM CdCl_2_. The ROS production was determined using Muse Cell Analyzer. Whole-cell lysates were then analyzed with Western blot using indicated antibodies. Actin was used as a loading control. All results are presented as means ± SE of three experiments. * *p* < 0.05 vs. control.

**Figure 3 molecules-27-05215-f003:**
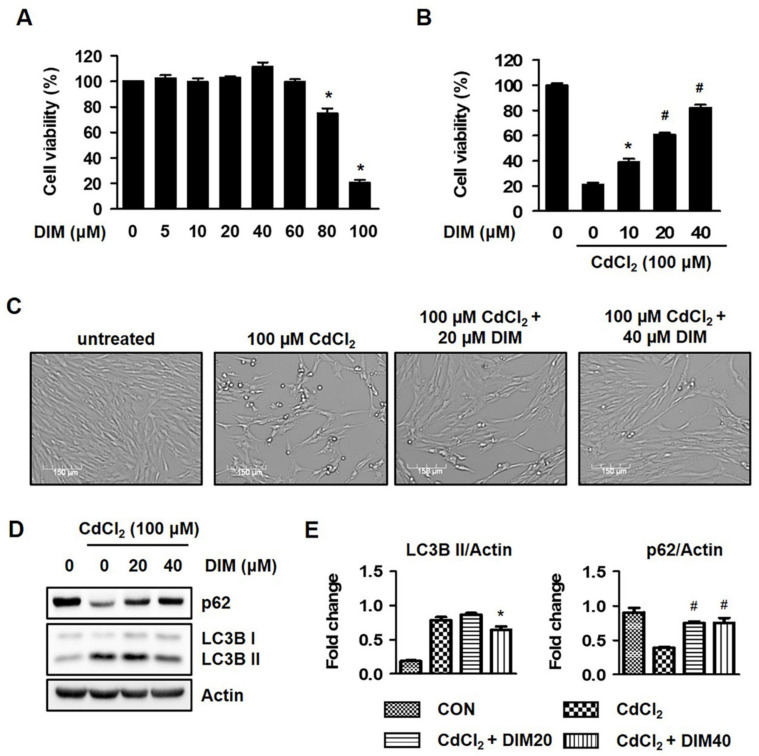
DIM inhibited Cd-induced cytotoxicity in HEL299 cells. (**A**–**C**) Cells were treated with different concentrations of DIM in presence or absence of 100 μM CdCl_2_ for 24 h; cell viability was measured using an MTT assay and microscopy (×400 magnification) as described in the Materials and Methods Section. (**D**,**E**) Cells were treated with DIM (20 and 40 μM) in presence or absence of 100 μM CdCl_2_ for 24 h, and whole-cell lysates were then analyzed with Western blot analysis. Actin was used as a loading control. All results are presented as means ± SE of three experiments. * *p* < 0.05 vs. control; # *p* < 0.001 vs. control.

**Figure 4 molecules-27-05215-f004:**
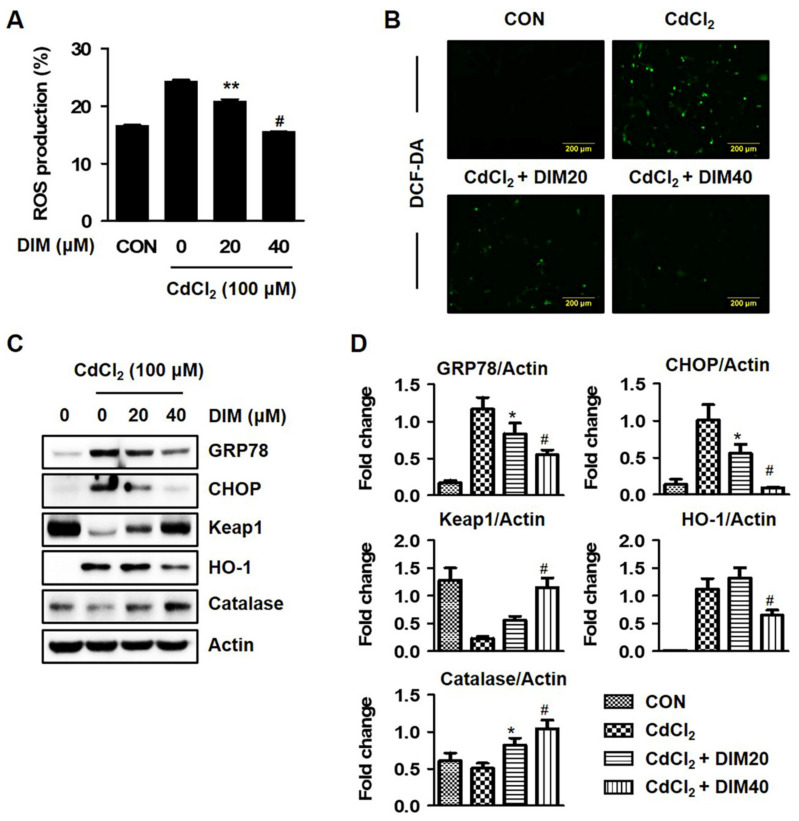
Effects of DIM on Cd-induced oxidative stress in HEL299 cells. (**A**,**B**) Cells were treated with the indicated concentration of DIM in presence or absence of 100 μM CdCl_2_ for 24 h, and ROS production was determined using fluorescence microscopy (×400 magnification) and Muse Cell Analyzer with Muse Oxidative Stress Reagent as described in the Materials and Methods Section. (**C**,**D**) Cells were treated with DIM (20, and 40 μM) in presence or absence of 100 μM CdCl_2_ for 24 h, and whole-cell lysates were then analyzed with Western blot analysis. Actin was used as a loading control. All results are presented as means ± SE of three experiments. * *p* < 0.05 vs. control; ** *p* < 0.01 vs. control; # *p* < 0.001 vs. control.

## Data Availability

Data supporting the findings of this paper are available from the corresponding author upon reasonable request.

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
