# Peer review of "Diindolylmethane Inhibits Cadmium-Induced Autophagic Cell Death via Regulation of Oxidative Stress in HEL299 Human Lung Fibroblasts"

_molecules, 2022, doi:10.3390/molecules27165215_

Round 1

Reviewer 1 Report

The work of Yeon-Seop Jung wt al. aims to check whether 3,3'-diindolylmethane (DIM) may have a protective effect on pulmonary fibroblasts exposed to cadmium. Cadmium has been shown to be a multi-organ damaging element, and because it is present in the environment, it seems to be worth considering how Cd-mediated autophagy-lysosomal degradation and cellular organelles homeostasis in various cell types can be acted upon.

The work is structured correctly. In the introduction, it would be possible to describe the molecular pathways of cadmium action, but the authors conclude that the experience will be contimulated. The methodology is brief but contains the most important information about the experiment.

The results are presented exhaustively, but I think that collecting all the data on one board is a bit illegible. Perhaps it would be worthwhile to arrange the figures so as to suggest to the Reader the order in which to interpret the results.

I believe the work is interesting and should be continued.
After minor corrections, I recommend the article for publication

Reviewer 2 Report

The paper by Jung et al describes the role of DIM as a therapeutic potential against oxidative stress Cd-induced.

The results obtained are not particularly new. It is known that Cd can induce oxidative stress. Regarding DIM, the authors themselves report, citing the literature, that it has antioxidant properties. Given these assumptions, the results obtained were predictable.

The figures should be described more in the results. The authors absolutely say nothing about figures 3 E and F, nor about 4D

The legends of the figures need to be more detailed.

Reviewer 3 Report

The authors claimed that CdCl2 induced autophagic cell death in HEL lung fibroblasts and the treatment of 3,3’-diindolylmethane (DIM) abolished the cytotoxicity of CdCl2. The experimental designs may not clearly prove the conclusions. Major points are listed as followed.

1. Although the authors observed the autophagy occurrence, it was unclear if the induced autophagy was cytotoxic or cytoprotective. The authors should use autophagy inhibitors, such as 3-Methyladenine or chloroquine, to demonstrate that blocking autophagy improves cell survival when exposed to CdCl2.

2. There were no legends of all the figures which makes it hard to review the manuscript. For example, what "CE" or "NE" means in Figure 2C?

3. For a demonstration of autophagy, the authors need to provide the punta formation of LC3 using a fluorescent protein-tagged LC3. The authors should also demonstrate the autophagic flux by GFP-RFP-LC3 construct to understand the cellular response of CdCl2. Due to the reduction of p62 (Figure 2C), the occurrence of autophagic flux will be expected.

4. Both NAC and AZO displayed the inhibition of CdCl2-induced ROS production (Figure 2E). The authors should provide the cell survival data of these treatments.

Round 2

Reviewer 3 Report

Although the authors have done some revisions, the data of cell proliferation in CdCl2 treatment under NAC (ROS inhibitor) or AZO 130 (ER stress inhibitor) condition should be provided to further demonstrate the involvement of ROS or ER stress.
